# Investigation of Microcrystalline Silicon Thin Film Fabricated by Magnetron Sputtering and Copper-Induced Crystallization for Photovoltaic Applications †

**Omid Shekoofa [1,2,]*** [iD]**, Jian Wang [1,]*, Dejie Li [1] and Yi Luo [1]**

1    Department of Electronic Engineering, Information Science and Technology National Laboratory, Tsinghua University, Beijing 100084, China; lidj@tsinghua.edu.cn (D.L.); luoy@tsinghua.edu.cn (Y.L.)
2    Satellite Research Institute, Iranian Space Research Center, Tehran 1997994313, Iran
*    Correspondence: o.shekoofa@isrc.ac.ir (O.S.); wangjian@tsinghua.edu.cn (J.W.)
†    This paper is an extended version of paper published in the 26th Iranian Conference on Electrical Engineering (ICEE), O. Shekoofa and J. Wang, Fabrication of P-Type Microcrystalline Silicon Thin Film by Magnetron Sputtering and Copper Induced Crystallization, Mashhad, Iran, 2018, pp. 289–293.

**Abstract:** Microcrystalline silicon, which is widely used in the microelectronics industry, is usually fabricated by chemical vapor deposition techniques. In recent years, magnetron sputtering has been considered as an alternative because it is a simpler, cheaper and more eco-friendly technique. The big drawback of this technique, however, is the need to recrystallize the as-deposited amorphous silicon, which can be done by metal-induced crystallization. Among the different suitable metals, copper has not been extensively investigated for this purpose. Furthermore, the applicability of the microcrystalline film prepared by this method has not been evaluated for photovoltaic device fabrication. Therefore, this paper reports the fabrication of p-type microcrystalline silicon thin film by magnetron sputtering and copper-induced crystallization techniques, and evaluates its appropriateness for solar cell fabrication. In the first step, 60 nm of silicon followed by 10 nm of copper were deposited on n-type silicon wafer and glass substrates, both by the magnetron sputtering technique. Then, the as-deposited samples were annealed at temperatures from 450 °C to 950 °C. The crystal properties of the resulting films were characterized by Raman and X-ray diffraction spectroscopies and optical and secondary emission microscopies, while their electrical characteristics were determined by Hall-effect, J-V curve and external quantum efficiency measurements. These characterizations confirmed the formation of a layer of microcrystalline silicon mostly in the <111> direction with a crystallization ratio of 93% and a largest grain size of 20 nm. The hole concentration and mobility of the fabricated p-type microcrystalline silicon layer were about $10^{17} \sim 10^{19}$ cm$^{-3}$ and 8 cm$^2$/V.s, respectively. By using the fabricated film as the emitter layer of a p-n junction solar cell, a good rectification ratio of 4100 and reverse saturation current density of 85 nA.cm$^{-2}$ were measured under dark conditions. The highest photovoltaic conversion efficiency, i.e., 2.6%, with an open-circuit voltage of 440 mV and short-circuit current density of 16.7 mA/cm$^2$, were measured under AM1.5G irradiance. These results indicate that microcrystalline silicon created by magnetron sputtering and copper-induced crystallization has considerable potential for photovoltaic device fabrication.

**Keywords:** microcrystalline silicon thin film; magnetron sputtering; copper-induced crystallization; photovoltaic

## 1. Introduction

High quality poly-crystalline silicon (poly-Si) is in great demand in the electronics industry, due to its low cost, the need for less semiconductor materials and simpler fabrication compared with monocrystalline Silicon (C-Si). The main applications of poly-Si are electronics (such as thin film transistors (TFTs) used in flat-panel, liquid-crystal displays), optoelectronics (optical Blu-ray discs) and photovoltaic (PV) devices (thin film solar cells) [1–4]. Poly-Si films may be fabricated by variety methods that result in different crystal sizes and properties. Nowadays, plasma-enhanced chemical vapor deposition (PECVD) is the most widely used method for poly-Si thin film fabrication [5–8]. However, this technique requires a large volume of $SiH_4$ gas for the deposition of Si as a raw material; the majority of the $SiH_4$ gas is screened out by the vacuum system, reducing the Si deposition rate to lower than 10% [9,10]. In addition, $SiH_4$ is a highly flammable, toxic and not eco-friendly gas which requires many safety considerations and precautions for its storage, transportation and exhaust from the operational PECVD system. In contrast to PECVD systems, magnetron sputtering (MS), as a physical vapor deposition (PVD) technique, generally uses inert gases such as argon as the reaction gas for the creation of plasma, and solid target materials as the source of the deposited film, making it a much simpler and safer approach than chemical vapor depositions (CVDs). Additionally, the MS technique has a higher raw material utilization (~30%) for Si deposition in comparison to CVD processes. Another important advantage of MS systems is the elimination of the need to use toxic or flammable gases. Furthermore, MS is a scalable deposition technique which is suitable for the fabrication of large-scale devices while maintaining the uniformity of the deposited layer; the technique makes it possible to easily deposit silicon and other semiconductors onto various rigid and/or flexible substrates with different properties such as glass, wafer, metal, polymers, fabrics, etc. Therefore, MS has many advantages and much potential as an eco-friendly, low-cost technique for the mass production of Si thin film solar cells [11–14].

However, the fabrication of such solar cells by MS has not been widely investigated; just a few research groups have reported successful approaches. There are two main reasons for this: The first is the very short effective diffusion length of the photo-generated carriers in a-Si made by MS (which prevents them from passing the p-n junction of the solar cell and reaching the electrode contacts); the second is the large number of dangling bonds in the sputtered a-Si thin film which act as traps for photo-generated carriers [15,16].

One effective solution to these problems is the crystallization of a-Si and its conversion to micro- or poly-Si by thermal annealing. This process combines the dangling bonds of a-Si film and forms orderly covalent bonding which reduces the trapping of the photo-generated carriers and increases the short circuit current density in the thin film solar cell. Usually, a pure solid-phase crystallization of a-Si only happens at very high temperatures, which limits the applicability of crystallized a-Si for the fabrication of solar cells on low-cost substrates such as ordinary float glass. Using the metal-induced crystallization (MIC) method can reduce the activation energy and temperature for a-Si crystallization to a much lower percentage of the eutectic point of Si-metal binary systems [17,18].

Among different metals, aluminum has been widely studied [19–21], and different methods of aluminum-induced crystallization (AIC) have been proposed [22–25]. Another candidate element for MIC is copper which, unlike Al, has not been widely investigated for this purpose; indeed, just a few research papers have reported satisfactory results regarding polycrystalline film fabrication by copper-induced crystallization (CuIC) [26–28].

Accordingly, in this research, we not only investigate the fabrication of micro-Si thin film by MS deposition and CuIC techniques, but also evaluate its appropriateness for PV applications. We study the impact of the thermal annealing conditions on the quality of the fabricated micro-Si layer by different characterization methods. Then, using the prepared p-type micro-Si thin film as the emitter layer, a p-n junction solar cell is fabricated over an n-type Si wafer, and the electrical and its PV properties are assessed to show whether such a layer is suitable for the fabrication of low-cost thin film solar cells.

The structure of this paper is as follows. In Section 2, the fabrication process of a micro-Si thin film by CuIC is explained in detail. Section 3 presents the list of characterization tests and measurements which are conducted to reveal the properties of the fabricated thin films. The results of these characterizations are discussed and analyzed in Section 4. In the next section, and based on the acquired insight from study of the CuIC process, the principles and mechanisms of CuIC, when it converts the a-Si film to micro-Si layer, are explained and illustrated graphically. Finally, a summary and highlights of the conducted research are presented in Section 6.

## 2. Micro-Si Thin Film Fabrication Process

The step by step process of the fabrication of a micro-Si thin film crystallized by copper induction is shown in Figure 1. In the first step, a 60-nm layer of a-Si was deposited on two types of substrate by the direct current (DC) magnetron sputtering of a p-type Si target at a power level of 250 W for 180 s. The first type of substrate was a high-quality one-side polished monocrystalline n-type Si wafer, made by the Czochralski (CZ) method with a <100> crystal orientation, a 2-inch diameter, a thickness of 300 μm, and a resistivity of 10 Ω.cm. The second substrate was a group of quartz glass pieces with dimensions of $10 \times 10$ mm$^2$. The samples made on the first type of substrate were later used for electrical and PV characterizations of the fabricated thin films, while the second type of samples were used for material characterization purposes.

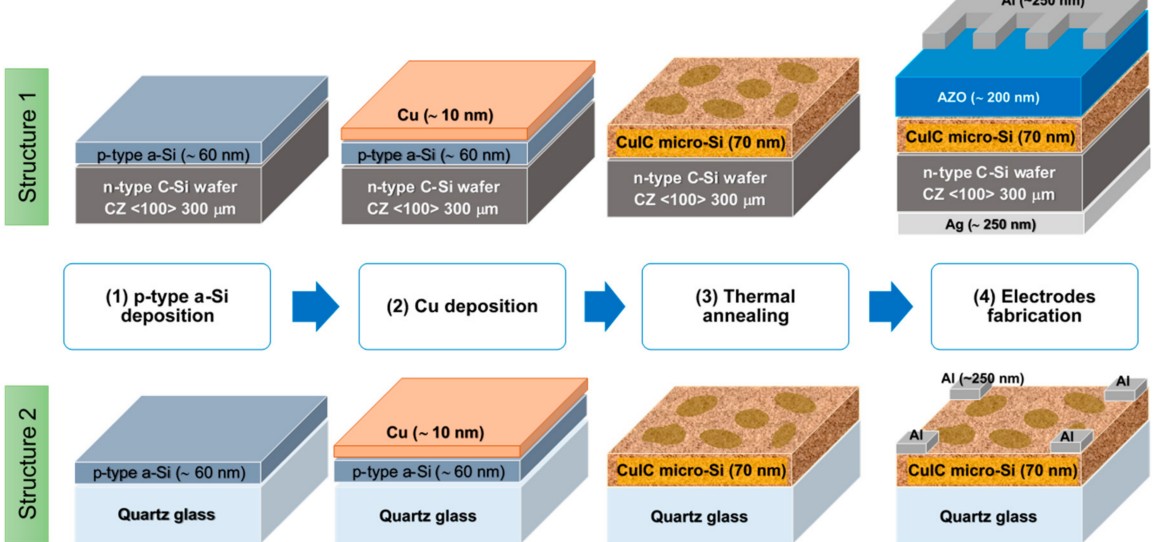

**Figure 1.** Step by step process of CuIC micro-Si thin film fabrication in this research: (top row) deposited p-Si/Cu/AZO tandem layers on n-type C-Si wafer, and (bottom row) deposited p-Si/Cu tandem layers on quartz glass.

Following the deposition of the a-Si layer, 10 nm of Cu was deposited by DC sputtering at a power of 200 W for 30 s in the same round of sputtering without breaking the vacuum of the sputtering chamber. The substrates were heated to 170 °C using the sample holder heater of the sputtering machine. The as-deposited samples after this step are shown in Figure 2a. We also deposited 200 nm of aluminum zinc oxide (AZO) over the Cu layer (for a selected group of samples with a C-Si wafer as the substrate). This transparent conductive layer was formed by the sputtering of AZO that lasted for 250 s at a DC power of 250 W (Figure 2b). Consequently, two structures of (n-type C-Si/p-type a-Si/Cu/AZO) and (glass/p-type a-Si/Cu) were created, as shown in Figure 1.

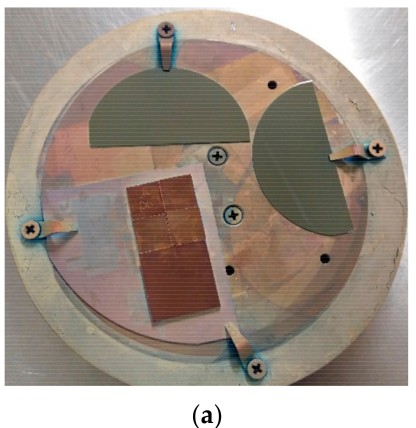
(**a**)

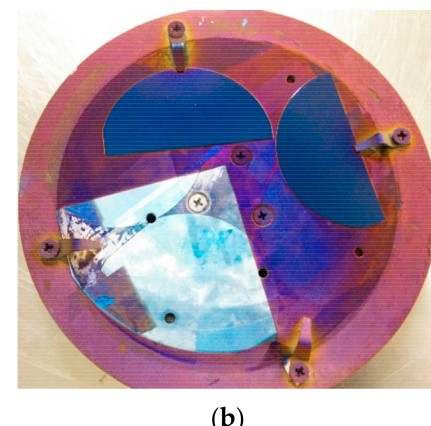
(**b**)

**Figure 2.** (**a**) the as-deposited thin film samples, (**b**) after AZO deposition.

Prior to loading the substrates (i.e., n-type C-Si wafers and quartz glass) inside the sputtering chamber, they were cleaned using a three-step procedure employing carbon tetrachloride/ acetone/ethanol in an ultrasonic cleaning machine; each step of the cleaning process lasted 10 min. To remove the native oxide layer from the surface of the n-type C-Si wafers, they were dipped into a buffered hydrofluoric acid (HF) solution for 30 s. Finally, all the samples were washed with deionized water and dried with high purity nitrogen blow to avoid the risk of contamination which can drastically reduce the quality of the final fabricated product.

All the aforementioned deposition processes were carried out with an MSP3200P magnetron sputtering machine under the conditions listed in Table 1. The base vacuum pressure was set to $5 \times 10^{-4}$ Pa, while it increased to 2 Pa and 1.2 Pa during the sputtering of Cu and Si respectively, by flowing Ar gas at a rate of 40 sccm. We used a p-type Si target with a diameter of 80 mm, a purity of 99.9999% and a resistivity of 0.01 $\Omega$.cm. The Cu and AZO targets had the same diameter as that of the Si target, but with much lower purity, i.e., 99.99%.

**Table 1.** Parameters of Deposition of Cu and Si by Magnetron Sputtering.

| MS Parameters | Cu | p-Type Si | AZO |
|---|---|---|---|
| Base Pressure [Pa] | $< 5 \times 10^{-4}$ | $< 5 \times 10^{-4}$ | $< 5 \times 10^{-4}$ |
| Sputtering Pressure [Pa] | 2 | 1.2 | 2 |
| Sputtering DC Power [W] | 200 | 250 | 250 |
| Duration [sec] | 30 | 180 | 250 |
| Deposition Rate [nm/sec] | 0.33 | 0.33 | 0.8 |
| Thickness [nm] | 10 | 60 | 200 |
| Ar Flow Rate [sccm] | 40 | 40 | 40 |
| Substrates Temperature [°C] | 170 | 170 | 170 |

When the deposition process was complete (and the tandem layers of Si and Cu were laminated by AZO), the fabricated samples were cut into several pieces with approximate dimensions of 1 cm × 1 cm. In the next step, the deposited thin films went under thermal annealing treatment. This was done in a FURNACE-1200 electrical annealing system equipped with a quartz vacuum tube which can heat the samples to 1200 °C. As shown in Figure 3, several thermal treatments were applied to the fabricated samples in an $N_2$ environment (to prevent oxidization) kept at a pressure of ~10 Pa. The applied annealing temperature was increased from room temperature to the desired crystallization temperature (450 to 950 °C) at a rate of 10 °C/min. The annealing process continued at the defined

temperature for between 2 and 4 h. After the annealing process was complete, the samples were cooled to room temperature via natural cooling.

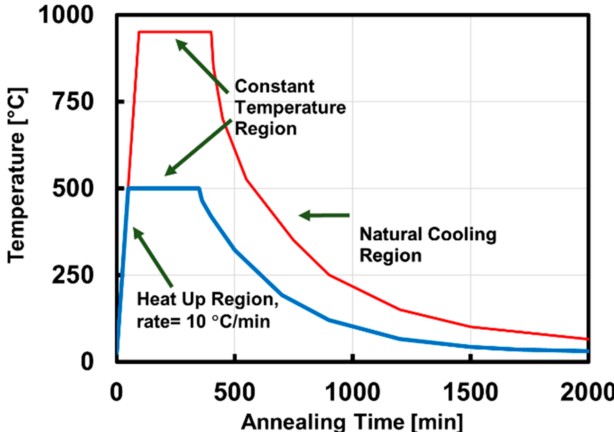

**Figure 3.** Profile of the thermal treatments applied in the third step of the CuIC thin film fabrication process.

In order to prepare the annealed samples for characterization tests and measurements, metal electrodes were created on the front side of the micro-Si layer by sputtering 250 nm of aluminum at room temperature. For the samples made on a C-Si substrate, the front electrode bus bar and finger patterns were formed using stencil masks, while for the samples on glass substrates, four metal electrode pads with sizes of $2 \times 2$ mm$^2$ were created on the four corners of each sample (for Hall effect measurements). In the last step of the device fabrication, the backside of the samples were completely covered by a layer of silver with a thickness of 250 nm, made by DC sputtering (Figure 1). In the next section, the process by which to study and analyze the fabricated thin film will be introduced.

## 3. Thin Film Characterizations

A series of measurements and characterization tests were performed on the fabricated micro-Si thin films in order to analyze their material and electrical properties. Raman spectroscopy and X-ray diffraction (XRD) technique were used to study the crystallization rate of the microcrystalline thin films. Raman peak shift was measured for the wave numbers from 400 to 600 cm$^{-1}$ with a laser wavelength of 532 nm by a LabRam HR spectrometer, made by HORIBA. The crystal orientation and grain size of the crystals formed in the micro-Si layer were studied by XRD spectroscopy. Then, XRD spectra was measured by a Rigaku SmartLab system equipped with a Bruker D8 diffractometer using CuK$\alpha$ radiation ($\lambda\alpha1 = 1.5406$ Å), covering 2-$\theta$ angles from 10° to 90°. We studied the surface morphology of the thin film samples on a glass substrate using an Olympus BH3-WHP6 optical microscope. A Merlin Field Emission Secondary Electron Microscopy (FE-SEM) system made by ZEISS was used to study the morphology and observe the microcrystal layer formation for the samples made on the C-Si substrate. Furthermore, we used Energy Dispersive Spectroscopy (EDS) measurements to determine the elemental and chemical properties of the micro-Si films. The EDS system was made by "OXFORD INSTRUMENTS", equipped with an X-MaxN Si drift detector with a detection area of 50 mm$^2$. The electrical properties of the fabricated samples were studied by Hall Effect, External Quantum efficiency (EQE) and J-V curve measurements. Study of the carrier concentration and mobility was conducted by Hall Effect measurements. We used an HMS 3000 system made by ECOPIA for this purpose. EQE measurement of the fabricated samples on C-Si wafers was performed using a QEX10 system made by "PV Measurement Inc." for a wavelength range of 300 to 1100 nm. Finally, the J-V characteristics of the fabricated p-n junction were measured using a Keithley-2400 source meter in dark and illuminated conditions. The illumination condition was adjusted in accordance with the AM1.5G irradiance standard by means of an XES-70S1 solar simulator.

## 4. Results and Discussions

### 4.1. Material and Crystal Properties

The impact of thermal annealing temperature on the crystallization ratio of the fabricated films was studied by Raman spectroscopy in the first step of characterization process. Figure 2 shows the normalized results of Raman peak shift due to annealing temperature variation from 450 to 900 °C. The Raman peak of C-Si, located at 520 cm$^{-1}$, is also shown as the reference. The Raman spectra of the samples annealed at 450 to 550 °C showed a broad peak at around 520 cm$^{-1}$ with high levels of intensity offset for wavenumbers lower than 480 cm$^{-1}$ and higher than 530 cm$^{-1}$, which means that a large portion of the fabricated films are not crystallized under these annealing conditions, but that part of the a-Si layer is converted into nano-Si and/or micro-Si. With raising the annealing temperature to 700–750 °C, narrower peaks appeared closer to 520 cm$^{-1}$. However, the intensity offset of the measured spectra was still much higher than that of C-Si in the entire off-peak range. The existence of this offset implies that a considerable part of the film was not fully crystallized, and was probably composed of a mixture of a-Si and nano- and micro-Si. Finally, when another sample was annealed at 900 °C, a very narrow peak at 518.8 cm$^{-1}$ was detected, but even at this high temperature, a small peak was observed at around 500 cm$^{-1}$ (as shown by a dashed oval in Figure 4a) which confirmed that the crystallization process had not run to completion.

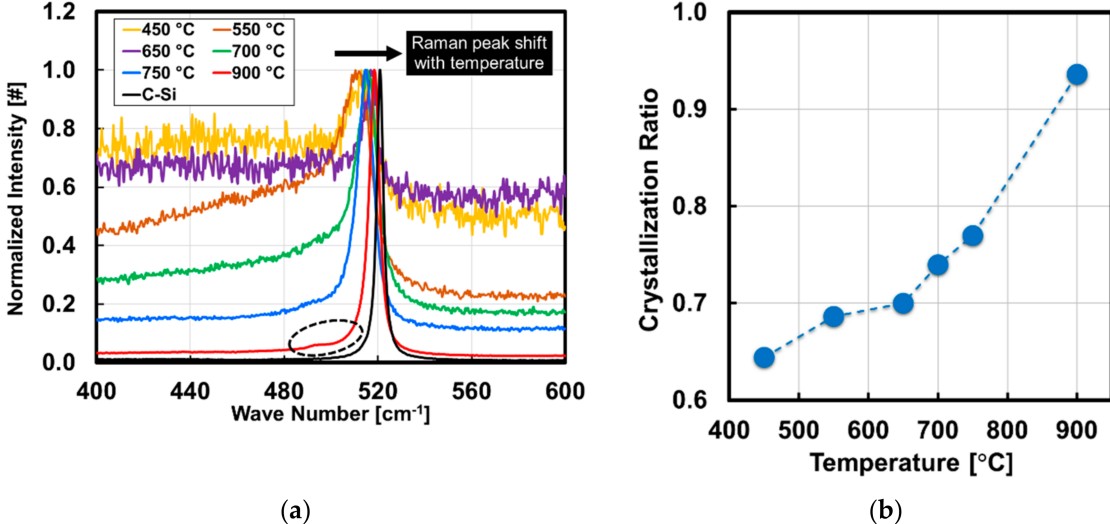

**Figure 4.** (**a**) Normalized Raman spectra of the CuIC micro-Si films fabricated on a glass substrate after thermal annealing at various temperatures, (**b**) The crystalized volume fraction of the fabricated CuIC micro-Si films vs. annealing temperatures.

To calculate the crystallized volume fraction of the annealed samples, we used Equation (1), which determines the crystallization ratio ($R_x$) based on the Raman spectra intensity of different phases of Si crystallization in the thin film layers.

$$R_x = \frac{I_c + I_m}{I_c + I_m + 0.85 \times I_a} \tag{1}$$

In this equation, $I_a$ and $I_c$ are the intensity of Raman spectra at 480 cm$^{-1}$ and 520 cm$^{-1}$, which correspond to the a-Si and C-Si phases, respectively, while $I_m$ is the average intensity of Raman spectra from 494 to 507 cm$^{-1}$, which represents the intensity of the mixture of a-, nano- and micro-Si phases [29]. As shown in the graph of Figure 3, for annealing temperatures lower than 800 °C, which is the eutectic point of a Si-Cu binary alloy system, $R_x$ was less than 80%. More precisely, we reached a crystallization ratio of $R_x \approx 77\%$ at 775 °C, which is in accordance with the result of a similar experiment carried out

in [30]. The highest values, i.e., $R_x$ was 93.6%, were calculated from the Raman spectra of the sample annealed at 900 °C.

In the second step of the characterization process, we investigated the crystal orientation and calculated the grain size of the CuIC micro-Si films by XRD spectroscopy. The results of four samples annealed at 550, 650, 750 and 950 °C are shown in Figure 5. The micro-Si crystals in the sample annealed at 550 °C were not dominant in any orientation, except for a very small peak in the <220> plane. After annealing the thin film samples at higher temperatures, i.e., 650 °C and 750 °C, three peaks, i.e., Si <111>, <220> and <311>, appeared in the XRD spectra. It was observed that the Si <111> peak was stronger than the <220> and <311> peaks, which supports the explanations given in [31]. When we annealed the sample at a much higher temperature, i.e., 950 °C, the corresponding Si <111> peak got narrower and stronger, which indicates that the crystal growth in the <111> direction is dominant and with a higher preference. However, at the same time, the peaks of <220> and <311> become narrower and stronger, so the relative intensity of $I_{<111>}/(I_{<220>} + I_{<311>})$ reduced from 1.175 to 1.08. This means that increased thermal treatment has a negative impact on the fabricated film properties, because it increases crystal growth in undesirable orientations and can reduce the uniformity and quality of the resulting microcrystal layer.

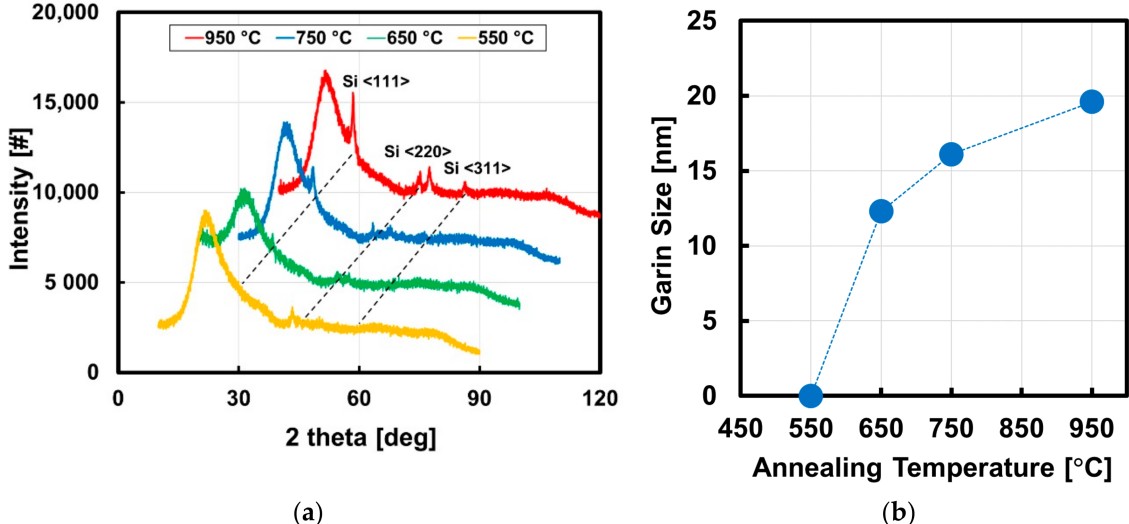

(a)                                     (b)

**Figure 5.** (**a**) XRD spectra of CuIC p-type micro-Si thin films fabricated on a quartz glass substrate. Note: the 2nd, 3rd and 4th XRD spectra are shifted by 10°, 20° and 30°, and by 2500, 5000 and 7500 units on the horizontal and vertical axes, respectively, for more clarity, (**b**): the calculated grain size of the CuIC micro-Si films formed in <111> plane vs. annealing temperature.

To determine the grain size of the micro-Si in <111> direction, we used the Scherrer equation which is introduced in Equation (2), wherein $\lambda$ is the wavelength of the scanning X-ray, $\theta$ is the half angle of diffraction peaks and $B$ is the half width of diffraction peak (in radians). The crystal grain sizes can be calculated from the full width at half the maximum (FWHM) value of the micro-Si thin film peak [29]. The result of the calculated grain size is shown in Figure 5b. The largest calculated grain size of the CuIC micro-Si thin films in <111> orientation was almost 20 nm. In addition, it was observed that for annealing temperatures lower than Si-Cu eutectic point, smaller grain sizes occurred.

$$D = \frac{0.89\lambda}{B \cos \theta} \tag{2}$$

Following the XRD measurements, the surface of the micro-Si thin film deposited on glass substrates (i.e., glass/p-Si/Cu samples) was studied using optical microscopy. Figure 6 shows three images, taken at three zoom levels, of the samples which was annealed at 550 °C. These optical micrographs reveal interesting details about the nucleation of the Si atoms, and the beginning of the

formation of a micro-Si layer due to the heat treatment. The left micrograph in Figure 6 shows the overall texture of the fabricated film, which is obviously neither uniform nor continuous. However, several regions may be distinguished as crystallization areas. The middle image shows more detail of the micro-Si film morphology of one of these areas, which contains some bulks of residual copper on the film surface after the thermal treatments. This leftover metal can reduce the light absorption efficiency in PV applications. A closer look at the surface of this sample shows there are many small pink spots (indicated by arrows (a)) on the film surface which are likely clusters of nano-Si nucleation centers that individually formed shortly after the beginning of annealing treatment at 550 °C. These clusters expanded and coalesced into each other as the annealing process progressed. According to Figure 6, several areas with dendrite shapes (indicated by contours (b)) formed and grew, leading to further lateral expansion and the creation of larger grain boundaries. There are also some dark spots on the film surface which could be small clusters of residual Cu after the thermal annealing step in the CuIC process.

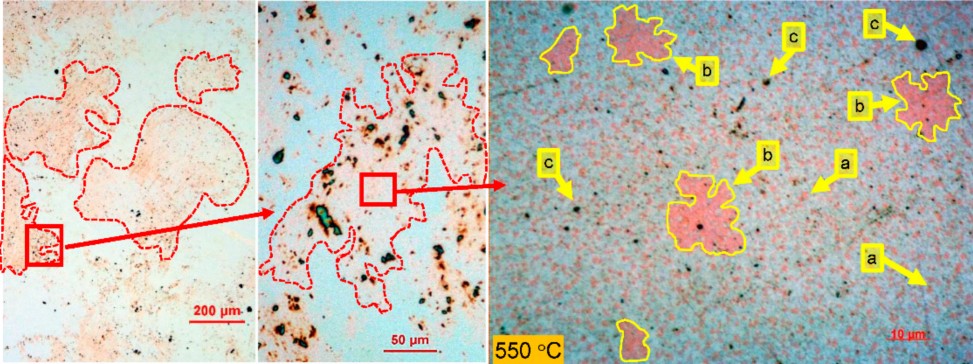

**Figure 6.** Optical micrograph of the CuIC micro-Si thin film with (glass/p-type Si/Cu) structure after annealing at 550 °C.

When we annealed another sample at 750 °C, we observed quite different crystallization conditions, as shown in Figure 7. The major part of the film surface was covered by a bright beige continuous layer, which may be seen in the background of this figure. According to the measured Raman spectra, we think that this area is mainly in the micro-Si phase.

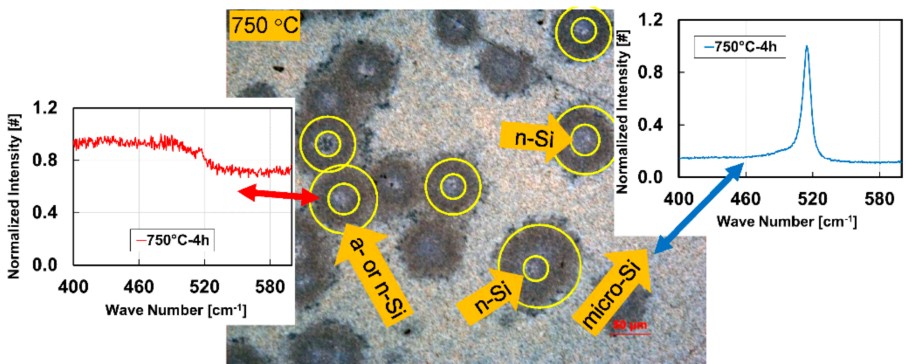

**Figure 7.** Optical micrograph of the CuIC micro-Si thin film with (glass/p-type Si/Cu) structure after annealing at 750 °C (the scale bar is 50 μm).

Firstly, much larger crystallized areas were observed on the film surface, which formed a bright beige, almost continuous layer. According to the measured Raman spectra (right-side graph), this area is mainly in the micro-Si phase (as indicated by the flash arrow labelled "micro-Si"). Inside this mico-Si layer, there are several dark disk areas, marked by yellow rings, with approximate outer diameters of 70–100 μm and inner diameters of 20–30 μm. Based on the measured Raman spectra, these areas

had crystallized yet, and were probably composed of mixture of a- and nano-Si phases. In the center of each disk, there is a small bright area surrounded by the inner circle, which was likely composed of the nano-Si phase. This area is increased in size (and the corresponding ring became narrower) as the crystallization process progressed, either by extending the annealing time or increasing the annealing temperature.

Figure 8 shows the surface of another sample annealed at 950 °C. At such a high temperature, several large crystallized areas with dimensions of 10–50 μm were created; their dendrite shapes are clearly observable (as indicated by red ovals). However, the created layer is no longer continuous, perhaps due to the low thickness of the deposited film. As mentioned before, at this high temperature, crystal growth happens in different orientations; this could be another reason for the poor quality of the thin film after annealing at 950 °C.

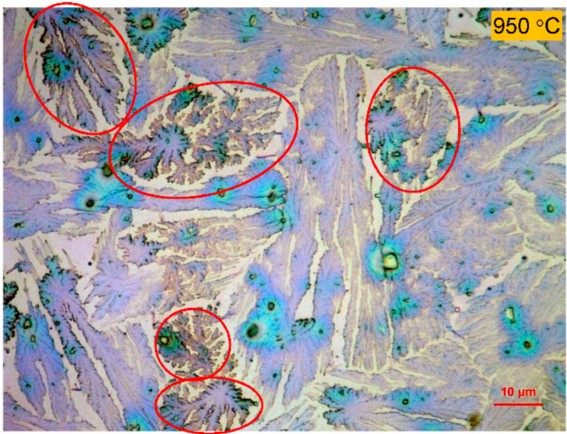

**Figure 8.** Optical micrograph of the CuIC micro-Si thin film with (glass/p-type Si/Cu) structure after annealing at 950 °C.

In the next step, the surface of the fabricated films was studied by SEM microscopy in order to gain a better insight into the nano- and micro- structures of the crystallized Si layer. The micrograph of the deposited (n-type C-Si/p-type micro-Si/Cu) film structure after annealing at 500 °C is shown in Figure 9a. According to this micrograph, the majority of the film surface was covered by a uniform mixture of Cu and Si in amorphous and nano-crystalline phases (e.g., the area labeled "a"). The brighter spots are likely the clusters of copper (indicated by "b") formed on the mixture (a + nano) of the phases of Si and Cu. This SEM image confirms the previous results that indicate the very low crystallization ratio of the film at low temperatures. Figure 9b displays the SEM of a sample annealed at 600 °C, which shows different crystallization conditions in comparison to the former sample. A fine and almost uniform layer which is still not fully crystallized formed in the background of the images. However, several dark areas are distinguishable, defined by yellow contours and labelled "c"; these are likely composed of the nano- and micro-Si phases which resulted from the applied thermal treatments. The residual bulk of Cu which did not contribute to the CuIC process is seen as bright spots (due to the SEM imaging nature) and was labelled "b". This SEM image confirms a small grain size i.e., ~20 nm, as determined by XRD measurement, shown in the graph of Figure 5b.

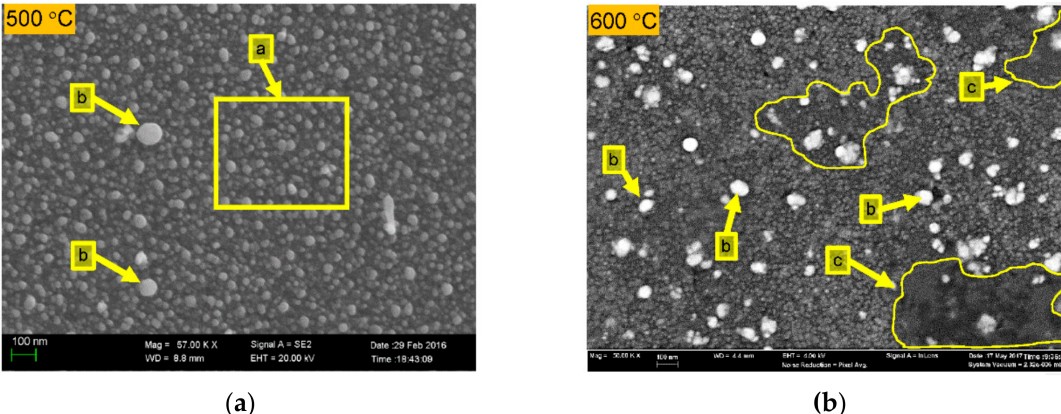

**Figure 9.** (**a**) SEM micrograph of CuIC micro-Si thin film with (n-type C-Si wafer/p-type Si/Cu) structure after annealing at 500 °C, (**b**) SEM micrograph of the CuIC micro-Si thin film after annealing at 600 °C.

*4.2. Electrical and Photovoltaic Properties*

As the next part of characterization process, the electrical properties of the fabricated CuIC micro-Si thin film were analyzed using Hall Effect measurements. To this end, the doping concentration and carrier mobility were determined for the samples annealed at different temperatures; see Figure 10. At an annealing temperature of 550 °C, the hole concentration was in the order of $10^{15}$ cm$^{-3}$, while the mobility parameter was extremely high. Therefore, such a micro-Si film is not suitable as the emitter layer of poly-Si solar cells, because according to [32,33], the typical values of the hole concentration and carrier mobility for an emitter layer should be in the range of from $10^{16}$ to $10^{19}$ cm$^{-3}$ and from 1 to 100 cm$^2$/V.s, respectively. Thermal treatment at a higher temperature, i.e., 750 °C, led to a considerable decrease in hole mobility, i.e., to 70 cm$^2$/V.s, which was in an appropriate range. At the same time, the opposite trend was observed for the hole concentrations of the fabricated film after annealing at 750 °C, which reached to more than $10^{18}$ cm$^{-3}$. When thermal annealing was conducted at 900 °C, the hole concentration and mobility reached 70 cm$^2$/V.s and $5 \times 10^{19}$ cm$^{-3}$, respectively. This behavior confirms that Cu atoms can play the role of p-dopants more effectively with increasing the thermal budget of the CuIC process.

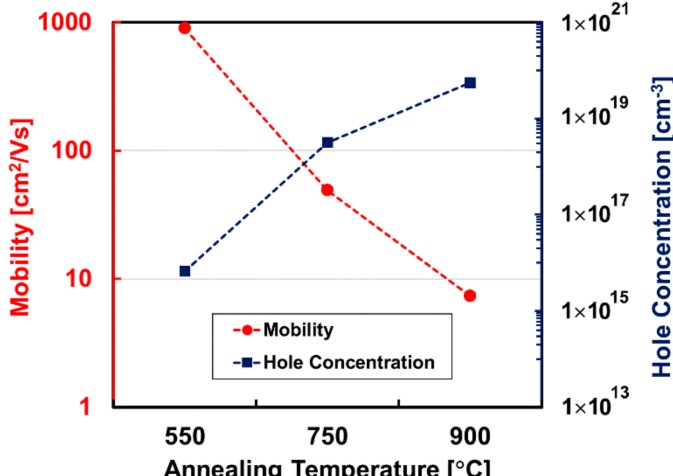

**Figure 10.** Variation of the hole concentration (blue) and mobility (red) with thermal annealing temperature.

When the material and electronic properties of the deposited thin films had been investigated, we fabricated a simple solar cell using the CuIC micro-Si thin films as the emitter layer and created

metal electrode contacts for the thin film samples fabricated on n-type C-Si wafers. The simplified structure of this device is shown in the inset of Figure 11a. In this device, CuIC micro-Si thin films form the p-side of the solar cell junction, while an n-type C-Si wafer forms the n-side. The AZO layer acts as an interface between the micro-Si layer and metal electrode contacts, and reduces the series resistance of the final solar cell. Figure 11 shows the Rectification Ratio (RR), dynamic resistance and J-V characteristics of the solar cell made by the CuIC thin film layer which was fabricated using the optimized process. RR is defined as the ratio of the forward bias current of the p-n junction to the current at the same bias voltage but with reverse polarity. The variation of RR with temperature is shown in the blue curve of the graph in Figure 11. Under the dark condition, the created p-n junction showed good rectification behavior, with the highest $RR = 4100$ at $\pm 1$ V and a reverse saturation current of $J_0 \approx 85$ nA.cm$^{-2}$. Then, we calculated the dynamic resistance ($R_d = \Delta V/\Delta J$) of the p-n junction over a bias voltage of $\pm 1$ V in order to determine the series resistance ($R_S = R_d|_{J_{cell} = 0}$) and shunt resistance ($R_{Sh} = R_d|_{V_{cell} = 0}$) of the fabricated solar cell. As shown in Figure 11, the values of $R_S = 14.0$ $\Omega$.cm$^2$ and $R_{Sh} = 63.7$ $\Omega$.cm$^2$ were too high and too low, respectively. These results, in line with the material and morphological characteristics, imply that the quality of the fabricated thin film was not sufficient for use in photovoltaic applications. However, this represented a significant improvement in comparison to our previous results reported in [34].

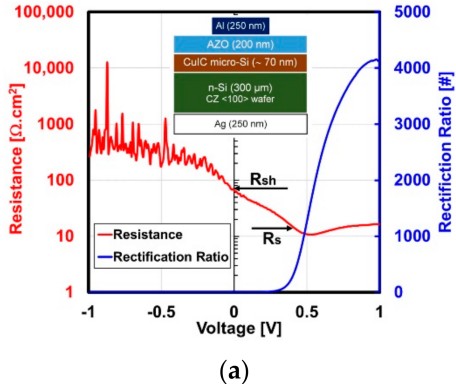 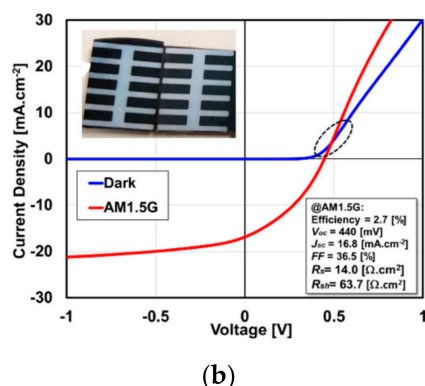

(**a**)    (**b**)

**Figure 11.** (**a**): the p-n junction rectification ratio and cell dynamic resistance, (**b**) the J-V curve of the best sample of a micro-Si solar cell fabricated by MS and CuIC. The structure of the designed solar cell and two fabricated samples are shown in the inset pictures.

Under standard illumination of AM1.5G irradiation, the fabricated solar cell showed a significant PV effect, with an open circuit voltage of $V_{oc} = 440$ mV, and a short circuit current density of $J_{sc} = 16.7$ mA/cm$^2$ for the best prepared sample. The PV energy conversion efficiency was limited to 2.7% and the Fill Factor (FF) to 36.5%. Finally, we observed a cross-over voltage effect at $V_{bias} = 488$ mV (indicated as a dashed oval), which is an intrinsic property of thin film solar cells, wherein the under-light J-V curve intersects wtih the dark J-V curve.

As the final step of the characterization tests, the EQE spectral response of the CuIC solar cells samples was measured. The results of the EQE measurements of the two samples which were fabricated from thin films annealed at 850 °C and 900 °C are shown Figure 12. According to this figure, these solar cells only had relatively good response in the green band of the spectrum; beyond 650 nm, their response especially in the red region of the spectrum was very low, which limited $J_{sc}$ @ 550 nm to only 16 mA·cm$^{-2}$. Therefore, the photo-generation in this device was restricted to a narrow band of the whole spectrum, reducing the overall $J_{sc}$. Additionally, the measured energy band gap ($E_g$) was 1.304 eV, while the typical $E_g$ of poly-Si solar cells made by aluminum-induced crystallization (AIC) is lower than 1.2 eV [35].

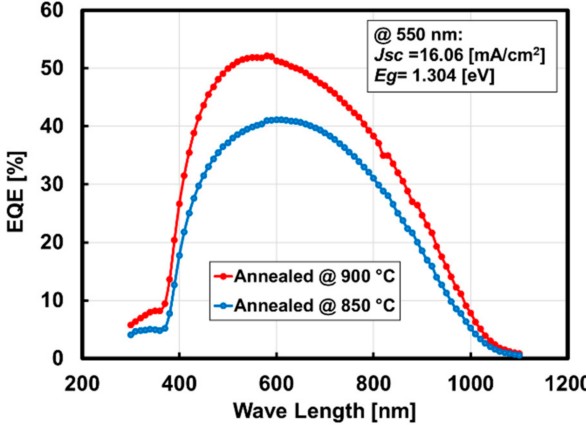

**Figure 12.** EQE responses of the solar cells fabricated by CuIC micro-Si thin film.

## 5. Principle and Mechanism of CuIC

According to the conducted research, and based on the results of characterization tests, we here describe and explain the process of CuIC of a-Si in three main steps, as illustrated in the top row of Figure 13. In the first step, the deposited a-Cu and a-Si layers start to mix (forming a $Cu_3Si$ compound) in several local centers of their interface, initiated from centers where the effective surface of the contact between Cu-Si is larger. The existence of $Cu_3Si$ was verified by EDS scanning of the surface of a sample annealed at a temperature close to Cu-Si eutectic point.

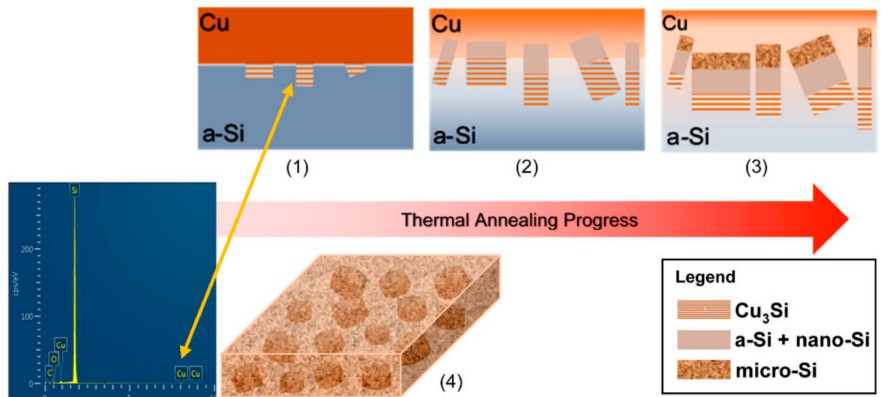

**Figure 13.** Illustration of the principle and mechanism of micro-Si film fabrication by CuIC.

Then, in the second step, the formation of $Cu_3Si$ increases with extending the thermal annealing time or increasing the annealing temperature. These $Cu_3Si$ bulks migrate into a-Si layer-like ribbons (or needles), leading to the crystallization of the a-Si layer as a mixture of nano- and micro-Si ribbons which become longer and bigger. As mentioned above, the micro-Si cannot start to grow from everywhere, and cannot grow in all directions; it can only evolve into a-Si layer within a ~100 micron radius area from the $Cu_3Si$ ribbons, as observed in Figure 7 [28]. In fact, studying the lateral crystal growth for CuIC of glass/a-Si/Cu in [31] revealed that the majority of the micro-Si ribbons had <110> orientations with respect to the a-Si film surface normal, and that they grew in the <111> and <211> directions. Therefore, the migration of $Cu_3Si$ ribbons mainly takes place perpendicular to the interface plane between the Cu and Si layers.

The crystal orientation can be explained by the thermal strain energy generated in the micro-Si film due to differences in the thermal expansion coefficients between a-Si and a-Cu, as well as a-Si and glass (or C-Si wafer). By extending the thermal annealing time of the CuIC process, micro-Si growth in the <111> direction becomes dominant due to the lower energy needed for crystal formation in this

orientation. Furthermore, since $Cu_3Si$ has a vacancy in its crystal band structure, it can passivate the a-Si films in a similar way to a-Si passivation by hydrogenation.

## 6. Conclusions

In this research, a p-type micro-Si thin film was fabricated by MS and CuIC techniques, and its characteristics were measured and analyzed in order to evaluate its suitability for photovoltaic applications. The micro-Si film was created by the deposition of a-Si/a-Cu tandem layers by the DC sputtering method. The prepared samples were annealed at different temperatures, wherein the Cu atoms acted as the inducing metal to reduce the crystallization temperature. The material and crystal properties of the fabricated samples were studied by Raman and XRD spectroscopies. It was confirmed that micro-Si films were formed in the <111> plane with a grain size of ~20 nm and crystallization ratio of $\approx$ 93%. The morphology of the micro-Si films was observed to evaluate the quality of the thin film and to understand the mechanism of a-Si crystallization by CuIC. Measurement of the Hall effect showed that Cu can act as a p-dopant, and the hole concentration and mobility of the created mico-Si film may be controlled by the thermal annealing time and temperature. Then, the prepared micro-Si film was used as the emitter layer of a solar cell when it was deposited on a heavily doped n-type Si wafer. The fabricated p-n junction possessed good rectification characteristics under dark conditions, i.e., $RR = 4100$ and $J_0 \approx 85$ nA.cm$^{-2}$. Under AM1.5 G standard irradiation, the fabricated solar cell showed a noticeable PV effect, with an efficiency of $\eta = 2.7\%$, an open circuit voltage of $V_{oc} = 440$ mV and a short circuit current density of $J_{sc} = 16.7$ mA/cm$^2$. To our knowledge, this is the most comprehensive demonstration of solar cell fabrication by the CuIC method. Therefore, the proposed method is expected to open up a new path for the fabrication of low-cost, eco-friendly, thin-film Si solar cells. However, the use of such a micro-Si thin film as the emitter layer of a p$^+$-n junction solar cell, and the observation of promising PV performance, were only possible after annealing the samples at high temperatures (>750 °C); their PV efficiency was not comparable to that of the solar cells made by AIC, for example. Indeed, a big difference between the temperature at which the CuIC started and the temperature where the PV effect became noticeable was observed. The need for high temperature annealing indicates that Cu does not act as a p-dopant well enough at temperatures lower than 750 °C. This high annealing temperature limits the potential of such a micro-Si layer for the fabrication of thin film solar cells on low-cost substrates. The main drawbacks of these CuIC solar cells are their low $R_S$, $FF$ and $J_{sc}$, which were probably due to the high rate of carrier recombination in the p-type micro-Si (the emitter layer of the solar cell). Accordingly, it is necessary to look for a better metal that can be used for the MIC. However, considering that the whole fabrication process can be done in just three steps, we believe that it is interesting and has some potential for further development and optimization.

In summary, the highlights of this research are as follows:

- P-type, micro-Si, thin film preparation by MS and CuIC techniques, as a simple, low-cost, scalable and eco-friendly method for the fabrication of polycrystalline silicon.
- Demonstration of p$^+$-n junction thin film solar cell fabrication in only three steps, with an emitter layer made by MS and CuIC without any chemical process.
- Measuring a photovoltaic conversion efficiency of $\eta = 2.7\%$, an open circuit voltage of $V_{oc} = 440$ mV and a short circuit current density of $J_{sc} = 16.7$ mA/cm$^2$.
- Presenting a simple model for the CuIC principle, and illustrating the mechanism of micro-Si thin film growth during thermal annealing.

**Author Contributions:** Conceptualization, D.L. and J.W.; methodology, D.L., J.W. and O.S.; validation, O.S., J.W. and D.L.; formal analysis, O.S. and J.W.; investigation, O.S. and J.W.; resources, J.W. and Y.L.; writing—original draft preparation, O.S.; writing—review and editing, O.S., J.W., D.L. and Y.L.; visualization, O.S.; supervision, J.W. and Y.L.; project administration, Y.L.; funding acquisition, Y.L. All authors have read and agreed to the published version of the manuscript.

**Funding:** This work was supported in part by National Key Research and Development Program of China (2017 YFA0205800); National Basic Research Program of China (2015 CB351900); National Natural Science Foundation

of China (61176059, 61210014, 61321004, 61307024, 61574082, 51561165012); High Technology Research and Development Program of China (2015 AA017101); Tsinghua University Initiative Scientific Research Program (20161080068, 20161080062).

**Conflicts of Interest:** The authors declare no conflict of interest

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
