# Peer review of "Investigation of Microcrystalline Silicon Thin Film Fabricated by Magnetron Sputtering and Copper-Induced Crystallization for Photovoltaic Applicationsâ€"

_applsci, doi:10.3390/app10186320_

Round 1
Reviewer 1 Report
The authors are reporting on microcrystalline silicon thin films fabricated by magnetron sputtering and copper induced crystallization for PV applications. The article is well written and well presented!
However, it is majorly a replica of the first two authors published work [O. Shekoofa and J. Wang, "Fabrication of P-Type Microcrystalline Silicon Thin Film by Magnetron Sputtering and Copper Induced Crystallization," Electrical Engineering (ICEE), Iranian Conference on, Mashhad, 2018, pp. 289-293, doi: 10.1109/ICEE.2018.8472708.].
The Main addition in the article is the hole effect measurement, which does not really provide that much of additional information, especially, considering the amount of the replicated work in this article including most of the figures.
Author Response
Reply to reviewer 1
Dear reviewer 1, thank you very much for your comments. As you mentioned, this manuscript is an extended version of our conference paper which is cited as Ref. [33]. However, with all due respect, we believe that the current version has provided enough new data, graphs, figures and explanations which make it enough different than the conference paper. The main differences, are as follows:
- Due to the limited pages, we couldn’t provide enough details about the experiment in the conference paper while in the current version the fabrication process is explained and illustrated in more details so that the experiment can be repeated by the interested reader.
- The graphs of figure (4) and (5) are updated in this revised version by including new experiment data and showing more details on each graph, which give more information to the reader and better support the final results (and make them look different than the relevant figures of the conference paper).
- The optical micrographs of figure (7) and (8) are new which contain very important information about CuIC process. Figure (7) shows the phenomenon of narrowing of the rings of a-Si and nano-Si mixtures by thermal annealing, figure (8) shows the impacts of the thermal treatment at very high temperature of 950 °
- Figure (9) is added to this version, which shows the SEM micrograph of the as-deposited sample has been added in this version to help the reader better understand the difference of the thin film surface morphology before and after the thermal annealing.
- The holes’ concentration and mobility graphs are added to the manuscript, and their variations by temperature have been explained.
- According to the updated graph of figure (11)-right The photovoltaic conversion efficiency of the best fabricated sample is increased from 2.1% to 2.7% (it has been improved by 28% in comparison to the conference paper). Also, the graph of figure (11)-left is added to the paper, which shows the rectification ratio (blue) and dynamic resistance of the p-n junction under reverse and forward bias voltages (from -1 to +1 volts) under dark condition. This information helps to determine the quality of the created p-n junction.
- The EQE measurement results are presented in the graph of figure (12) which didn’t exist in the previous paper.
- In the conference paper we couldn’t explain the principal and mechanism of CuIC process due to the limited pages. Now, we have explained the principal of CuIC in section 5, and its process is illustrated as shown in figure (13).
Reviewer 2 Report
The manuscript presents the “Investigation of the Microcrystalline Silicon Thin Film fabricated by Magnetron Sputtering and Copper-induced Crystallization for Photovoltaic Applications”.
The topic is interesting and it is adapt to this journal. The subject may attract some interest to the readers. In general, this manuscript is well organized and written, with comprehensive literature review, detailing the framework approach of the study, clearly stated methodology and nicely presented findings. The manuscript provides sufficient background information regarding the topic proposed.
Few more specific comments and recommendations:
- define all notations that is used where the concept appears first mentioned in the text, some of the abbreviations are not explained;
The abstract is too extensive. ABSTRACT should give a pertinent overview of the work. I suggest that the ABSTRACT provide the following elements: (1) Background: Place the question addressed in a broad context and highlight the purpose of the study; (2) Methods: Describe briefly the main methods or treatments applied; (3) Results: Summarize the article's main findings; and (4) Conclusions: Indicate the main conclusions or interpretations.
In the Introduction Section, please provide more general information on the importance of research in order to emphasize the state of the art also (first general information, then specific).
- explain how this paper differs from the related ones already published;
- the author can highlight the usefulness of the study in the practical applicability.
Usually, conclusions are supported by the analysis or by the discussions included in the main text of the paper. Please provide and highlight relevant aspects of your work so that they contain 3 to 5 short bullet points that convey the essential conclusions of your article.
Author Response
Reply to Reviewer 2:
Dear reviewer 2, thank you very much for your positive comments about the topic of the research and its report. We also very grateful for your valuable suggestions for improving the manuscript. We have done our best effort to apply your comments and suggestions, as listed below:
- We carefully checked the whole manuscript to find notations without pre-introduction, and we edited and explained all of them.
- The abstract was revised and shortened according to the given comments. We tried to consider all of your suggestions in the edited abstract.
- In the Introduction section we added several sentences to give more information about the importance of the research.
- As indicated in the paper, the main goals of this research are: (a) to use copper-induced crystallization method in order to (b) crystalize the amorphous Si thin films deposited by (c) magnetron sputtering technique, and to investigate the fabricated polycrystalline layer appropriateness for being used in (d) photovoltaic applications. Actually, in most of the research chemical vapor deposition techniques have been studied while in our research magnetron sputtering is used as a simpler and eco-friendlier technique for thin film deposition. Beside, many research has reported on metal-induced crystallization, however, copper has not been studied as much as aluminum or nickel. Furthermore, the other research has investigated the application of silicon thin films made by copper-induced crystallization for TFT and OLEDs while in this research we evaluate its application for photovoltaic devices fabrication.
- The usefulness of the study is more emphasized both in Introduction and Conclusion sections.
- The “Highlights” of the research are added to the Conclusion section.
Reviewer 3 Report
This paper by Shekoofa et al. reports on the growth and characterization of metal-induced crystallized Si for photovoltaics. The work is carefully done and the manuscript is clearly presented. It can be accepted after minor revision. Here are some comments:
- On p. 4 “According to the profiles of Figure (3), several thermal treatments were applied to the fabricated samples in an N2 environment (to prevent the samples from oxidization)”. Does nitrogen react with the samples, i.e. is there any sign of nitride formation?
- Could the deposition of Aluminum as an electrode produce Aluminum Silicide at the interface? If so, could this affect the electric measurements?
- Did the authors attempt any detailed comparison of the Scherrer formula (eq.(2)) results for the size of crystallites with the SEM results? This could be achieved by doing a statistical distribution of grain size from a SEM image taking about, for example, 200 grains in to consideration. By inspection of Fig. 9(left) I am suspicious that a bimodal growth mode may exist. This could be revealed by the statistical analysis.
Author Response
Reply to Reviewer 3:
Dear reviewer 3, thank you very much for your valuable and positive comments. Followings are the answers to the proposed questions:
- Thank you for noticing such detail. According to the data resulted from EDS measurement, we didn’t observe any nitrogen or silicon nitride compositions in the thin film samples. However, we didn’t inspect it by more accurate measurements such as XPS. We will study this item in our future research for sure.
- Thank you for noticing this issue. We think the answer is “No”. Since the Al contacts were created by the deposition of Al on the fabricated micro-Si film at the room temperature, there is no chance to form aluminum silicide in noticeable quantity. Besides, if such a process takes place at high temperatures the formation of Al-Si binary can help to recrystallize the remained a-Si areas beneath the electrode contacts via the mechanism of aluminum-induced crystallization. Since such created microcrystalline is inherently p-doped (due to the existence of Al) it has the same doping type as the CuIC micro-Si layer and doesn’t have a considerable negative impact on the electrical properties.
- Thank you again for suggesting such an interesting item. As far as we understood, your concern is to compare the grain size resulted from Scherrer formula (Eq.(2)) and the grain size in Figure (9)-left. Actually, we calculated the grain size by using of Eq. (2) for the annealed samples which showed the creation of micro-Si grains with the largest size of 20 nm. Figure (9)-left shows the surface morphology of the as-deposited sample (before annealing) therefore we didn’t evaluate the grain size for that sample. However, we analyzed the SEM micrographs of figure (9)-right which (is thermally annealed at 600 °C), by ImageJ software. According to this brief analysis, we found out that more than 80 % of the film surface is covered by microcrystalline Si grains with the diameters between 5 to 20 nm (as shown in the below graph). We think this primary analysis confirms the results of Eq.(2) and supports your concern about the inexistence of bimodal growth mode. However, we believe this item needs more study and evaluation at this stage and we will consider it in our future research.

Round 2
Reviewer 1 Report
Thank you for the added/modified figures, details, your response, and efforts. The new version of the manuscript is better and reduces the worries about duplications of published work. The article was and still of high quality presentation and I think it is in a better shape for publication now.
Author Response
Thank you for your review.